# Online Estimation of Ship Dimensions by Combining Images with AIS Reports

Zishuo Huang [1], Qinyou Hu [1,*], Lan Lu [2], Qiang Mei [3] and Chun Yang [1]

1   Merchant Marine College, Shanghai Maritime University, Shanghai 200135, China; 202040110005@stu.shmtu.edu.cn (Z.H.); chunyang@shmtu.edu.cn (C.Y.)
2   Hainan Research Institute, Nankai University, Haikou 570100, China; nklulan@163.com
3   Navigation College, Jimei University, Xiamen 361021, China; meiqiang@jmu.edu.cn
*   Correspondence: qyhu@shmtu.edu.cn; Tel.: +86-21-38282915

**Abstract:** Ship dimensions are an important component of static AIS information, and are a key factor in identifying the risks of ship collisions. We describe a method of extracting and correcting ship contour information using inland waterway surveillance video combined with AIS information that does not depend on ship dimension data. A lightweight object detection model was used to determine the ship's position in an image. Dynamic AIS information was included to produce multigroup control points, solve the optimal homography matrix, and create a transformation model to map image coordinates onto water surface coordinates. A semantic segmentation DeepLabV3+ model was used to determine ship contours from the images, and the actual dimensions of the ship contours were calculated using homography matrix transformation. The mAP of the proposed object detection model and the MIoU of the semantic segmentation model were 86.73% and 91.07%, respectively. The calculation error of the ship length and width were 5.8% and 7.4%, respectively. These statistics indicate that the proposed method rapidly and accurately detected target ships in images, and that the model estimated ship dimensions within a reasonable range.

**Keywords:** contour extraction; object detection; semantic segmentation; coordinate mapping





## 1. Introduction

In water traffic scenarios, ship collision avoidance needs to use an automatic identification system (AIS) or radar and other navigational equipment to obtain the movement information of ships, and the two must complement each other. Using AIS reports, it is often difficult to accurately calculate the distance of the ship from its surrounding objects without considering the shape of the ship. The AIS position is determined by the position of the GPS antenna, and the distance between the GPS antenna and the ship periphery can range from tens to hundreds of meters. For ship–pier collisions, it is necessary to consider the transverse distribution of ships in the river, and use the ship width data to calculate the collision probability. The ship collision risk index (CRI) is calculated through the distance closest point of approach (DCPA), the time closest point of approach (TCPA), and other indexes to represent the urgency of the ship collision at a micro level. In the process of calculating the CRI between two ships, an AIS-equipped ship is often shown as a triangle or rectangle with the transponder at the center [1]. There are dimensional attributes in the AIS static information to represent the length and width of the ship, but in practice, these AIS data are often unavailable. Table 1 shows the dimensions and numbers of ships passing through two inland waterway channels in China within a given time period. It is clear from the table that ship dimension information is unavailable for more than 60% of the vessels. Given the possible consequences of ship collisions, the problem of unavailable ship size data needs urgent resolution.

**Table 1.** Available information for ship dimensions.

| | Bulk Carrier | | | Tanker | | |
|---|---|---|---|---|---|---|
| | No. Observed | No. with Available Information | Ratio | No. Observed | No. with Available Information | Ratio |
| Jingzhou waterway | 100 | 38 | 0.38 | 50 | 19 | 0.38 |
| Yichang waterway | 100 | 33 | 0.33 | 50 | 26 | 0.52 |

Images have become more widely used for information extraction by computer vision technologies. An augmented reality system based on a fusion of AIS and advanced image processing technology can provide auxiliary information for early warning of navigation risks for autonomous surface vehicles (ASVs) [2]. Such system can also be used for traffic supervision that enables vessels to conform to navigation regulations in key navigable waters [3].

Combining visual data with AIS information enables the estimation of the size of specific ships in the image. Remote sensing or visible light images have frequently been used to extract the contours of target ships, create matching external rectangles or ellipses based on the contour shapes, and derive longitudinal and transverse ship dimension information [4]. Therefore, contour extraction is the basis of ship size estimation, which has been widely used in the transportation sector.

There have been many studies of ship contour extraction. In conventional contour extraction methods, edge detection based on image characteristics has been used to determine contours. Yan et al. [5] improved the Canny edge detection algorithm using a two-dimensional wavelet Gaussian function to calculate the partial derivative of the structural filter gradient amplitude, and adopt maximum inhibition and threshold filters for edge detection and connection for ships. Gu et al. [6] used a binary image gradient calculation for edge detection, and determined the minimum enclosing rectangle for ship contours. Zhu et al. [7] demonstrated a ship recognition method that used a predicted shape template to determine ship contours using the Otsu method, with peak density detection and column scanning as well as a conventional area averaging algorithm. Nie et al. [8] used a binarized normed gradient (BING) algorithm to predict the location of a ship in SAR images, and used an active contour algorithm to predict ship contours iteratively. Standard ship contour extraction methods are simple but are often unable to extract deep image information, and are only suitable for simple scenes.

Convolutional neural networks (CNN) are widely used for image feature extraction in deep learning applications in networks such as VGGNet [9], GoogleNet [10], Inception [11], and ResNet [12]. CNNs have important applications in semantic segmentation; multicategory target contours can be accurately segmented using pixel-level classification of images, and they have been used in many ship contour extraction applications. The fully convolutional network (FCN) was commonly used to extract ship contours by categorizing each pixel in a remote sensing image into the bow, hull, land, and sea [13]. Bovcon et al. [14] developed a deep encoder–decoder framework (the water obstacle separation and refinement network) for autonomous crewless ship navigation that extracted the contours of several ship targets. Ust et al. [15] introduced a scaffolding learning regime (SLR) that trained an obstacle detection segmentation network under weak supervision for individual ship contour extraction. Kelm et al. [16] trained a CNN to identify central pixels; the network recognized a part of an input image and calculated a rotation angle, and used the central pixel to describe the upcoming directional change in the contour. Deep learning methods that rely on training data are more accurate than other established methods, and are highly adaptable to different scenarios.

Remote sensing data of a particular area are not frequently updated, and vessels are densely distributed on inland waterways, making it difficult to accurately extract contours

from remote sensing images at any given point in time. Visible light images are generally made from a horizontal perspective, and this perspective is not particularly suitable for accurate ship size estimation. Targeting the problem of missing information of some ship dimensions in a waterway, this study innovatively proposes an intelligent identification method of ship dimensions based on a fusion of inland waterway monitoring overhead image and AIS information. The research scenario is shown in Figure 1, the important stretches of the upper reaches of the Yangtze River that have high marine traffic flow and density. The main contributions of this study are as follows:

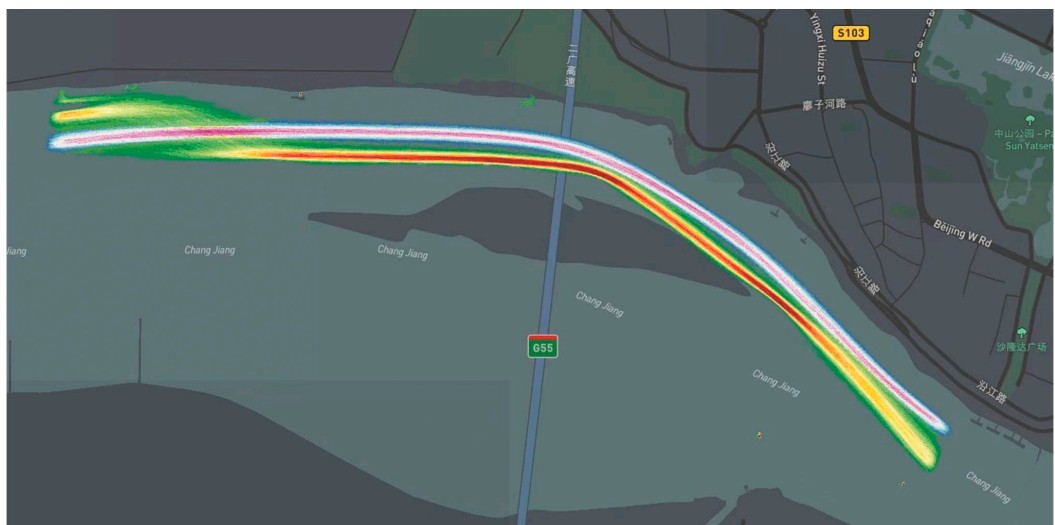

**Figure 1.** The traffic flow of Jingzhou Bridge in one month: white and yellow bands are used to represent the traffic flow in two different directions.

- A deep learning ship object detection model was developed based on a lightweight object detection model, and using the SENet attention mechanism to improve the network structure and increase the effectiveness of detection;
- An optimal homography matrix solution algorithm using several AIS control points was developed to determine the mapping relationship between image coordinates and water surface coordinates;
- Ship contours were extracted using the deep learning DeepLabV3+ semantic segmentation model, in conjunction with the homology matrix transformations to determine the real size of the vessel.

## 2. Related Studies

### 2.1. Ship Object Detection

Ship object detection technology has a research history of more than twenty years, and forms the basis for combining video and AIS information in our study [17]. The established methods include those based on the water–sky boundary, saliency detection, and moving object detection. Kim et al. [18] used a background algorithm to detect ships, and combined it with AIS to match ships with ship-related information. Fefilatyev et al. [19] developed a method using optimal water–sky boundary extraction combined with Gaussian distribution and the Hough transform. Yang et al. [20] designed a ship motion tracking system based on the FPGA that differed from traditional inter-frame difference methods, which had fixed frame intervals. Its inter-frame difference method was based on the adaptive extraction of key frames, and was used to adaptively detect ships moving at different speeds.

The CNN is used mainly to solve object detection problems, either in two-stage algorithms such as Faster RCNN [21] and Mask RCNN [22], or single-stage algorithms such as YOLO [23] and SSD [24]. Object detection algorithms that use deep learning overcome the shortcomings of target detection algorithms, such as lack of targeted region selection,

sliding window redundancy, and time complexity. Using a deep learning algorithm for ship object detection significantly improves detection. The rotational CNN algorithm [25] was used for text detection because of its excellent rotation detection capability that was introduced to ship target detection, and this produced good results. He et al. [26] combined the Gabor filter with the Faster RCNN to increase ship object detection accuracy from satellite images. Zhang et al. [27] preprocessed images with a support vector machine, and then processed the RoI images with a ship detection algorithm that used a regional CNN. This technique improved the recall and precision of small ship detection and the overall performance of the algorithm. Guo et al. [28] added rotation angle information to feature extraction, which increased the detection rate of ship objects at different scales, and greatly reduced the quantity of redundant information in the detection frame.

*2.2. Video Ranging Technology*

Video ranging technology is important in determining the true locations of imaged objects. The two types of video ranging are monocular ranging and binocular ranging. Monocular ranging has the benefits of a simple structure, rapid operation, and low cost; it is the main field of research at present, and a commonly used method is the Kalman filter (KF). Einhorn et al. [29] devised a feature-based extended Kalman filter (EKF) monocular visual ranging measurement algorithm that captured images with a single camera and used a depth estimation method to calculate a reliable initial estimate; the 3D positions were later reconstructed via an EKF. Chen et al. [30] introduced a monocular vision ranging measurement method based on pixel area and aspect ratio that predicted and optimized the pixel position in the subsequent frame using KF processing.

Another widely used technique was to calculate the distance to the object using object detection and camera projection. Raza et al. [31] used marker points to establish a line in the image, and used a linear equation to calculate the real-world distance between pixels based on the length of the line. Huang et al. [32] developed a monocular vision distance measurement method using object detection and segmentation; they developed a two-dimensional geometric vector model and used camera projection to calculate the distance. Zhe et al. [33] developed a monocular vision distance measurement method based on 3D detection, and created a regional distance geometric model to calculate the distance based on 3D detection and camera projection that produced good results when image detail was obscured.

The geometric principle of camera projection is shown in Figure 2, where $C$ indicates the fixed position of the camera; $A$ and $B$ are points on the target; $A'$ and $B'$ are the respective projections of points $A$ and $B$ on the plane of the camera sensor, which are recorded in the image; the projection line $AA'$ and camera optical axis plane belong to the surface method; $h$ is the height of the center of the camera sensor above the surface; $d$ and $d'$ are the surface distances from the target to the vertical axis of the center of the sensor; $\theta$ is the angle between the lines $OA$ and $OB$ on the horizontal surface; $f$ is the focal length of the camera. When the conditions $x = \frac{X}{2}$, $y < \frac{Y}{2}$, and $y = y'$ are satisfied, the distances between target points and the camera are calculated with the following:

$$d = \frac{f \cdot h}{Y/2 - y} \tag{1}$$

$$d' = \frac{d}{\cos \theta} \tag{2}$$

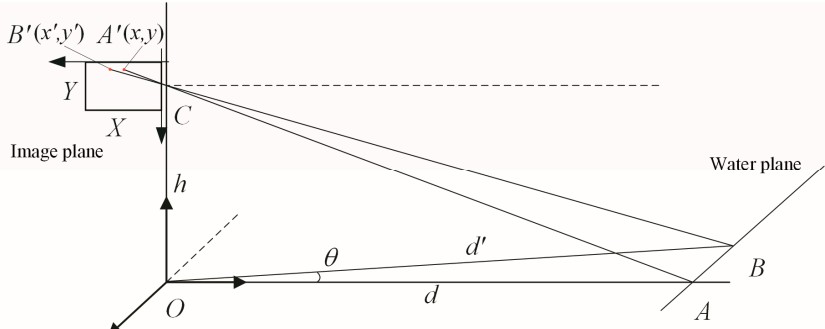

**Figure 2.** Geometric principle of camera projection.

## 3. Methods

There are four distinct stages of ship contour extraction: object detection, coordinate mapping, semantic segmentation, and image correction. Figure 3 is the technical roadmap of this research, and shows how data is transmitted between the various stages. In this section, we describe the key steps in detail.

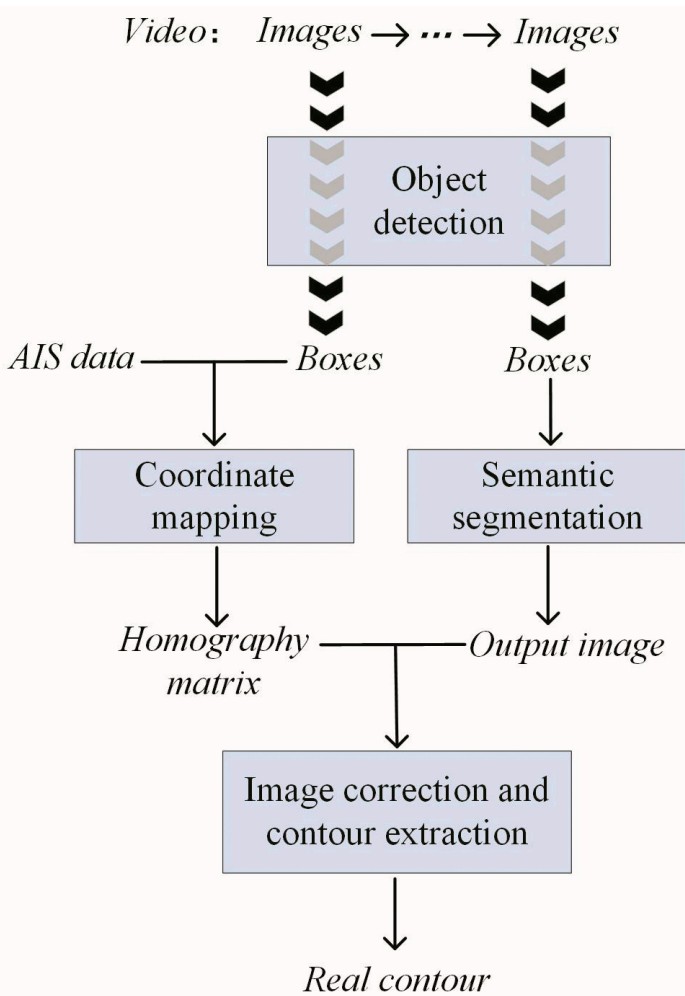

**Figure 3.** Technology roadmap of ship contour extraction and correction: video and AIS are input data, and the optimal homography matrix is first calculated, which is used to map the segmented image; after image correction and contour extraction, the final actual contour is output.

### 3.1. Object Detection Model

In general, object detection models often have a deep network structure, and require a large number of convolution layers with many parameters. Real-time inferences can only be performed if the devices they run on have adequate computing power. NanoDet is an excellent lightweight object detection model introduced in 2020 that dramatically reduces the number of parameters using a series of optimization methods. NanoDet can be quickly trained and ported to most embedded modules.

NanoDet uses several lightweight methods in the backbone, neck, and head, which enable it to balance accuracy, speed, and processing volume. The backbone is ShuffleNet V2, which removes the last layer of the convolution from the network and extracts 8, 16, and 32 downsampled features as the next inputs. ShuffleNet V2 is a CNN architecture that uses pointwise group convolutions to simplify the calculation of $1 \times 1$ convolutions, and uses channel shuffle to resist negative influences. The network greatly reduces computation, but maintains accuracy. The PAN module is a feature pyramid structure that performs upsampling and downsampling successively, which can fully integrate high-level features with low-level features. The neck is an optimized PAN that deletes all convolutions in the PAN, and only uses $1 \times 1$ convolutions extracted by the backbone for channel dimension alignment. An interpolation algorithm is used for upsampling and downsampling, and the multiscale feature map is added for feature fusion, which enables the network to learn the characteristics of multiscale targets. The FCOS is a typical anchor-free object detection algorithm with head detection through the neck of the output feature map pixel classification and bounding box regression to obtain the detection box. The optimized FCOS model was used as the detection head with abandoned weight sharing; it uses different convolutions to extract features at each layer and uses batch normalization, which uses deep separable convolution instead of group normalization. The number of convolution kernels and convolution channels also decreases, and the generalized focal loss function is used to resolve problems of convergence in training. In all, these methods greatly reduce redundant convolution and the number of parameters in the model, thus decreasing computation time.

SENet [34] is a spatial attention mechanism that increases the depth of a CNN and improves feature extraction. It consists of squeeze, excitation, and reweight functions. In the squeeze stage, the feature space with dimensions $c \times h \times w$ is compressed to $c \times 1 \times 1$ by global pooling, and the feature maps of a single channel are compressed into a weight factor. Two fully connected devices are used in the excitation stage. The first compresses the global information obtained from the global pooling; the feature dimension $c \times 1 \times 1$ is reduced to $c/r \times 1 \times 1$. The second fully connected device is used to map the feature back to $c \times 1 \times 1$ after ReLU activation. A sigmoid function is used to determine the normalized weight in a range of 0–1 of each channel to multiply the original feature map in the reweight stage.

The network structure is shown in Figure 4. SENet was added between ShuffleNetV2 and the optimized PAN with $40 \times 40$, $20 \times 20$, and $10 \times 10$ feature maps are input to emphasize useful features and suppress irrelevant features. Figure 4 shows the two feature maps before and after the SENet mechanism for comparison, and displays the SENet-enhanced features. After the calculations for PAN feature fusion and the FCOS detection head, the final output represents the locations of candidate boxes and their scores for different categories.

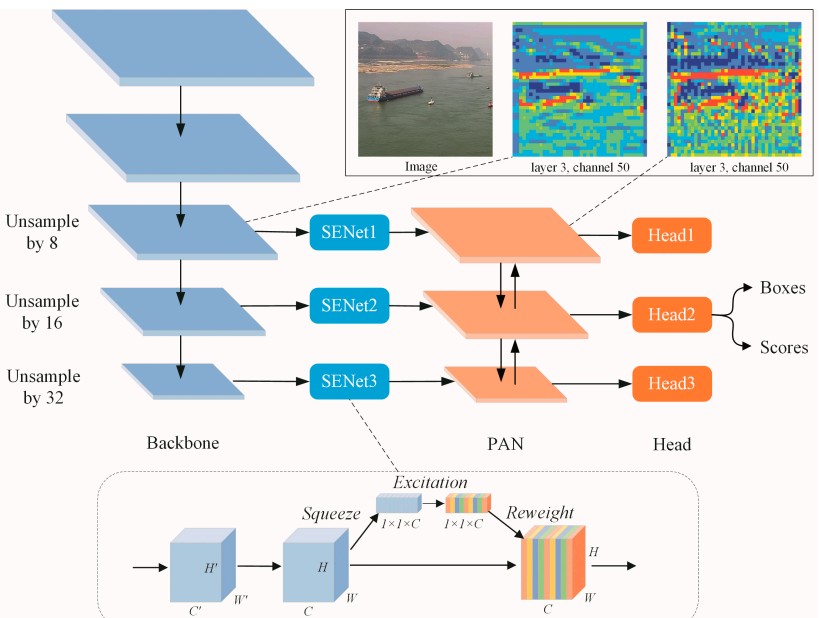

**Figure 4.** NanoDet–SENet network structure.

### 3.2. Coordinate Mapping

Mapping sensor image coordinates onto the water surface is key to matching image information with real-world information. When the camera attitude is constantly shifting, it is often necessary to combine optical ranging methods to ensure the camera view is parallel to the water surface. When parameters for camera height above the water surface, focal length, and pitch angle are combined with the projection equations, the depth map of the image can be calculated to predict the distance to each pixel on the water surface in the image. However, deploying surveillance cameras in inland river navigation areas is complex, and parameters such as height, focal length, and pitch angle are difficult to obtain in a timely manner. The use of data from fixed monitoring locations often requires using a homography transformation matrix to convert between the sensor image coordinate system and the water surface coordinate system, depending on the control points, and then mapping the pixel coordinates of the ship contour image to the water surface coordinates. The equation for the homography matrix transformation is as follows:

$$[x'\ y'\ w'] = [u\ v\ 1] \begin{bmatrix} a_{11}\ a_{12}\ a_{13} \\ a_{21}\ a_{22}\ a_{23} \\ a_{31}\ a_{32}\ 1 \end{bmatrix} = [u\ v\ 1]\boldsymbol{H} \tag{3}$$

where $u$ and $v$ are the pixel coordinates of the control points; the transformed coordinates are represented as $(u', v')$, where $u' = \frac{x'}{w'}$ and $v' = \frac{y'}{w'}$. $\boldsymbol{H}$ is the homography transformation matrix. At least four control points are required for the eight independent parameters in the solution of $\boldsymbol{H}$.

We used the real-time AIS position data and the corresponding observed ship positions in the image to create several control point coordinates. This necessitates that the camera be raised above the water's surface so that the hull takes up as much space as possible in the overhead image; on at least one channel, the camera can see the ship's side and back, as shown in Figure 5. The specific calculation steps of the optimal homographic transformation matrix are as follows.

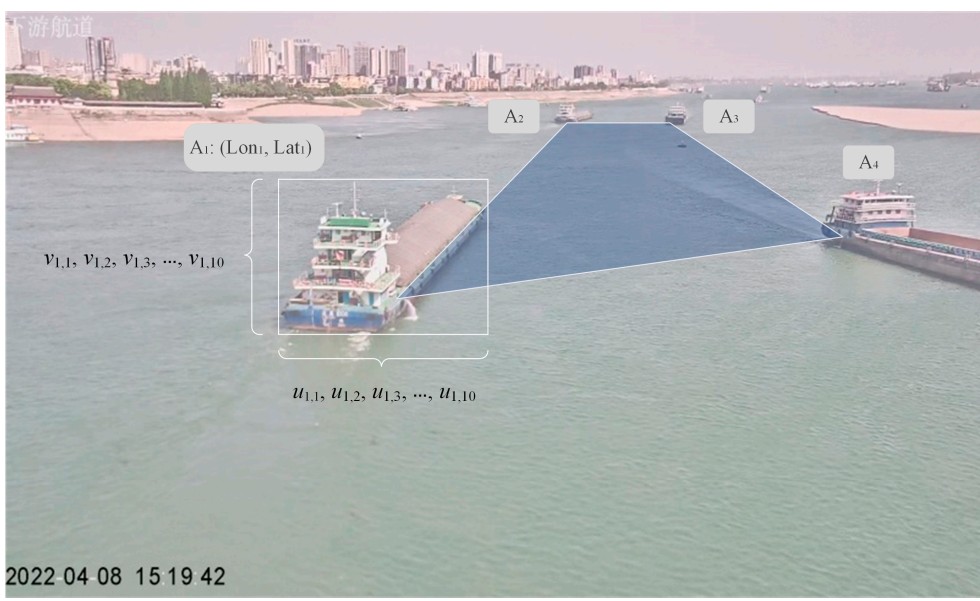

**Figure 5.** Multi-group control points constructed based on AIS information.

1.  Obtain the video and AIS data for two ships on different courses to form four groups of matching coordinates and AIS positions. One is $A_i = \{(Lon_i, Lat_i), Box_i\}$, where $Lon_i$ and $Lat_i$ are the latitude and longitude coordinates of the control point $i$, and $Box_i = \{(u_{i1}, v_{i1}), (u_{i2}, v_{i2}) \ldots, (u_{in}, v_{in})\}$ is the set of pixel coordinates of control points in group $i$, which consists of coordinates of $n$ equally spaced pixels. The control points of two ships traveling in the forward and backward directions are combined in pairs to produce several homography transformation matrices;

2.  Detect the key point $p_i$ of $Box_i$, where $p_i$ is the intersection of the extended side in the lateral and inferior directions from the saliency detection image produced by the LC model that obtains the saliency value of a pixel by calculating the sum of the distance in color between the pixel and all other pixels in the image. The two waterlines are determined with linear fitting;

3.  Select the two sets of coordinates for one ship, $Box_1$ and $Box_2$. Two key point pixels $p_1$ and $p_2$ are calculated, and Equation (3) is used to calculate the corresponding $n^2$ groups of water surface coordinates for $p_1$ and $p_2$. The error calculation of the homography matrix $H_j$ is as follows:

$$\vec{P_j} = p'_{2j} - p'_{1j} \, , j = 1, 2, 3, \ldots, n^2 \tag{4}$$

$$lon'_2 = lon_2 \times \cos\theta - lat_2 \times \sin\theta \tag{5}$$

$$lat'_2 = lat_2 \times \cos\theta + lon_2 \times \sin\theta \tag{6}$$

$$\vec{K} = (lon'_2, lat'_2) - (lon_1, lat_1) \tag{7}$$

$$\beta_j = \cos^{-1}\left(\frac{\vec{P_j} \cdot \vec{K}}{\left|\vec{P_j}\right| \times \left|\vec{K}\right|}\right), j = 1, 2, 3, \ldots, n^2 \tag{8}$$

where $p'_{1j}$ and $p'_{2j}$ are the mapped coordinates of $p_1$ and $p_2$, respectively, and $\theta$, the steering angle of the ship, is determined from the AIS information. When the smallest

$\beta_\alpha$ has been obtained, the matrix $\boldsymbol{H}_\alpha$ calculated by $\beta_\alpha$ is considered to be the optimal homography matrix.

### 3.3. Semantic Segmentation Model

Ship contour extraction requires segmentation of the area covered by the ship surface when viewed from above, but the area is often obscured by the superstructure of the vessel. Commonly used image segmentation algorithms are often greatly affected by noise and lack robustness, and it is difficult to determine the target area when it is obscured. However, the deep learning DeepLabV3+ semantic segmentation model [35] is highly accurate, robust, and not very susceptible to noise. Therefore, it is suitable for use in the segmentation of specific targets in a complex environment.

The network structure of DeepLabV3+ is shown in Figure 6. It consists of an encoder and a decoder. The main body of the encoder is a deep CNN with dilated convolution that controls the size of the receptive field by a rate ($r$) without changing the size of the feature graph. A greater value of $r$ produces a larger receptive field. The dilated convolution in the encoder is combined with a spatial pyramidal pooling module to produce multiscale information. The main constituents of the encoder are the following:

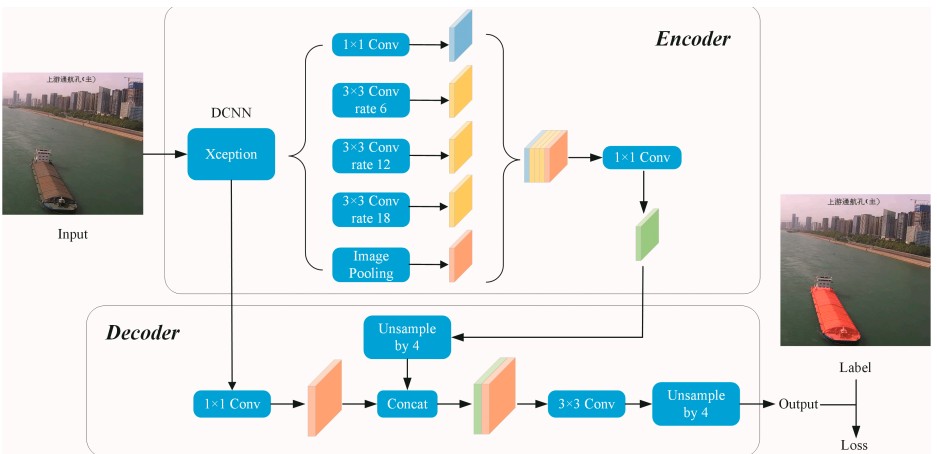

**Figure 6.** DeepLabV3+ network structure (The Non-English characters in the illustrations are meaningless).

*a.* One $1 \times 1$ convolution layer and three $3 \times 3$ empty convolution layers. The rate $r$ is (6, 12, 18) when the output step size is 16, and is doubled when the output step size is 8.

*b.* A global average pooling layer is used to produce image-level features that are then input into the $1 \times 1$ convolution layer and bilinearly interpolated to the original size.

*c.* Five features of different scales are combined in the dimension channel, and then input into the $1 \times 1$ convolution layer to be combined to produce 256 channels of new features.

The decoder can also combine low-level features with high-level features to increase segmentation accuracy. The main steps of feature fusion are as follows. The multiscale feature information is bilinearly interpolated and upsampled. The encoder then combines it with the original features that were extracted by the CNN. The combined feature information is then convoluted for simple feature combination. Finally, the combined features are bilinearly interpolated and upsampled to produce the segmentation results.

Xception [36] was used as the backbone network for feature extraction (Figure 6). Xception, which is an improved version of Inception V3, introduces depthwise convolution derived from Inception V3 to reduce model complexity and improve segmentation. The label in Figure 6 is from the annotation of the input images. The spatial pyramid module combined with dilated convolution combines multiscale image information. A larger value of $r$ will extract features from different regions of the image into a larger receptive field,

thus reducing obscuration by the superstructure. The marked area is the area where the ship is vertically mapped onto the water surface.

## 4. Results

### 4.1. Ship Object Detection

The experimental platform was a desktop computer with Windows 10, a GTX1050Ti GPU, and the PyTorch 1.8.0 framework. We created a coco dataset for training with 2535 images collected from surveillance videos from waterways upstream and downstream of the Yichang Yangtze River Bridge and the Jingzhou Yangtze River Bridge in China. The images included ships from different angles of different sizes, and in various lighting conditions. The image count was increased to 5070 using data enhancement methods such as noise processing, random angle rotation, random brightness adjustment, and simulated rain and fog weather conditions. Labelme software was used to annotate the images, which contained 12,376 ship objects altogether. The dataset was divided for training, validation, and testing in a ratio of 8:2:1. The input size for detection was $320 \times 320$. Stochastic gradient descent (SGD) was used for optimization. The initial learning rate was set to 0.01 because it is a common value suitable for most deep learning models.

The object detection models were trained and tested before and after validation, and their precision and recall were calculated. Precision represents the proportion of correctly predicted targets in total predictions, and recall represents the proportion of all target predictions that were correct. In general, precision decreases as recall increases.

To assess detection improvement attributed to SENet, P–R curves were plotted using precision and recall values for the two models before and after validation. In addition, the mainstream object detection algorithms Faster RCNN and YOLOv4, and the lightweight algorithm YOLOv4-Tiny using the test set, were selected for comparison to further assess the accuracy and efficiency of the NanoDet–SENet model detection. Table 2 shows the number and size of parameters for these three models. It can be seen from the table that NanoDet–SENet is an excellent lightweight model because its network complexity is much less than that of other models.

**Table 2.** Params number and size of different models.

|                | Total Params | Params Size |
|----------------|:------------:|:-----------:|
| Faster RCNN    | 137.08 M     | 522.91 MB   |
| YOLOv4         | 64.36 M      | 245.53 MB   |
| YOLOv4-Tiny    | 6.06 M       | 23.10 MB    |
| NanoDet–SENet  | **0.95** M   | **3.62** MB |

The P–R curves for the experiment are shown in Figure 7. Detection by several algorithms was assessed using four indicators: mean average precision (mAP), frames/s (FPS), precision, and recall, which are shown in Table 3. It can be seen from the table that the NanoDet–SENet model outperformed NanoDet, YOLOv4-Tiny, and Faster RCNN in terms of precision and the mAP, but did not perform as well as YOLOv4. The recall was greater than for NanoDet and YOLOv4-Tiny, but less than that for Faster RCNN and YOLOv4. The FPS was significantly greater than for all the other models except NanoDet. These results indicate that the SENet attention mechanism significantly influenced the detection effectiveness of the model, and that NanoDet–SENet detected objects almost as well as YOLOv4, although it is a simpler model.

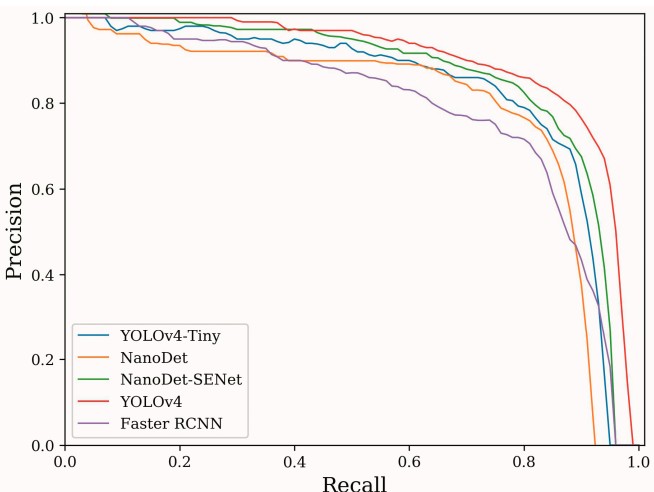

**Figure 7.** P–R curves for tested models.

**Table 3.** Comparison test results of target detection.

|  | Precision (%) | Recall (%) | mAP (%) | FPS |
|---|---|---|---|---|
| Faster RCNN | 69.68 | **81.51** | 78.84 | 6.35 |
| YOLOv4 | 86.23 | 79.64 | **90.20** | 9.10 |
| YOLOv4-Tiny | 84.20 | 74.55 | 83.68 | 30.71 |
| NanoDet | 78.96 | 74.72 | 76.55 | 38.62 |
| NanoDet–SENet | **85.09** | 76.83 | 86.73 | **37.47** |

### 4.2. Establishment of the Homography Transformation Model

We combined AIS information with video data to create the optimal homography transformation matrix between the sensor pixel coordinate system and the water surface coordinate system. The critical aspect of this algorithm is the synchronization of the time of the image sequence with the time of AIS information acquisition to ensure the accuracy of key point detection.

A Hikvision zoom network camera remotely captured video stream was transmitted using a real-time streaming protocol (RTSP), and a message queue telemetry transmission (MQTT) server was used to create an AIS information transmission platform. The video transmission rate was 3.2 Mbps with a 3.5 s delay, and the AIS signal delay was 6 s. As described in Section 3.2, the delays were eliminated, and the video and AIS information was used to obtain the coordinates of 250 sets of control points.

The key points were then tested. The accuracy of key point detection decreases for small-target ships, so we needed to ensure the ship detection boxes we selected as the control points were of adequate size and had well-defined contours, as shown in Figure 8. Figure 8a,b show ship objects detected at two distinct time points. The ship moving away from the sensor on the left side of the images was used to validate the optimal homography matrix. The red pixels in Figure 8c,d show the LC significant image pixels of change points, and the green pixels show key points after linear fitting. Figure 8d shows that the model predicted the exact position of the key point when the ship was further away from the sensor.

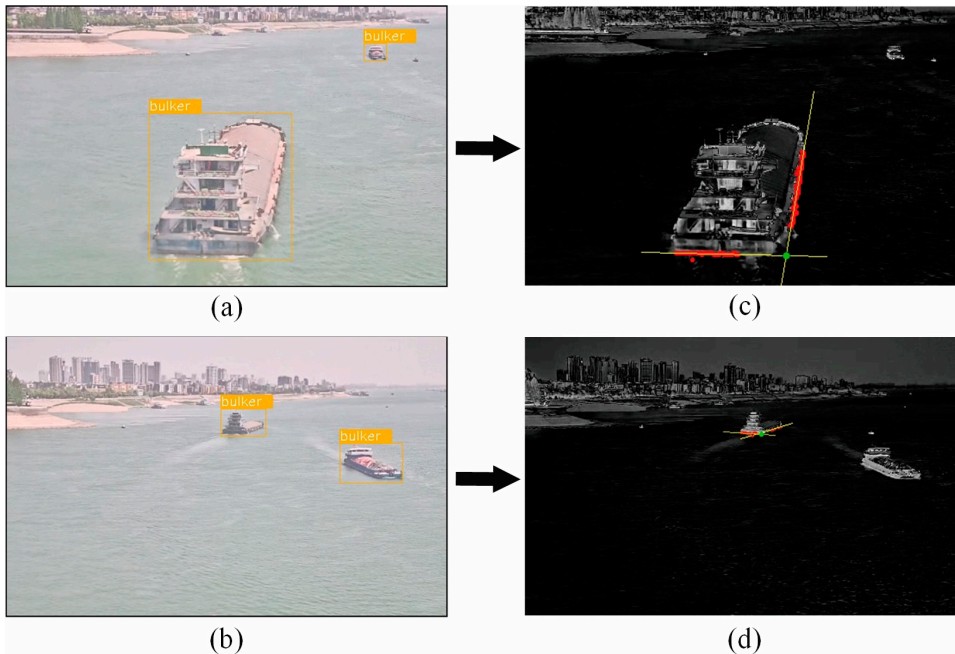

**Figure 8.** Key point detection progress: (**a**) ship object detection at first time, (**b**) ship object detection at second time, (**c**) key point detection at first time, and (**d**) key point detection at second time.

After we derived the optimal homography matrix, as described in Section 3.2, we used sensor data for the upstream and downstream directions to test the homography transformation model. Consecutive AIS coordinates were obtained, and the pixel coordinates of key points in the corresponding images were obtained and mapped onto the water coordinate system to predict the trajectory. Figure 9 shows that the trajectory of key points after coordinate mapping was very close to the AIS trajectory. The mean values of the distances between the corresponding points in the two sets of trajectories, 11.5 m and 77.8 m, were consistent with relative positions and distances between the real-world GPS antenna and the ship waterline inflection point. This result shows the accuracy of coordinate mapping using the homography transformation model.

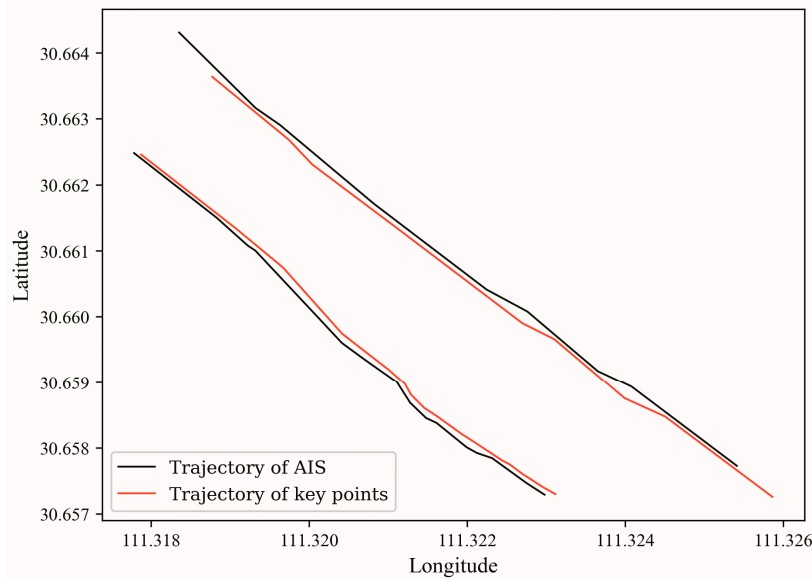

**Figure 9.** Test results of homography transformation model.

*4.3. Ship Contour Extraction*

Only images of bulk carriers and oil tankers were used, because the irregular shapes of passenger and container ships add unnecessary complexity to the development of the prototype model. We used surveillance cameras to obtain overhead views of vessels on the waterway to ensure that the deck surface matched the ship hull as much as possible. The object detection method was used to take automatic photos that were stored to create a semantic segmentation data set. The model training platform was a desktop computer with Windows 10 and an RTX3060Ti GPU using the Keras 2.2.5 framework. Target ship contours were labeled using Labelme software, and pixels were classified as either foreground or background. The ratio of the training set, validation set, and test set was 8:2:1. Online data enhancement was used to randomly amplify the image and label data in each batch in the training stage. The input data size for the model was $512 \times 512$.

The mean intersection over union (MIoU), which is a standard metric of the accuracy of a semantic segmentation algorithm, was used to assess the performance of the algorithm. The equation is as follows:

$$\text{MIoU} = \frac{1}{k+1} \sum_{i=0}^{k} \frac{p_{ii}}{\sum_{j=0}^{k} p_{ij} + \sum_{j=0}^{k} p_{ji} - p_{ii}} \qquad (9)$$

where $k$ is the number of pixel categories, $p_{ij}$ is the number of pixels that originally belonged to category $i$ but are predicted to be in category $j$, and MIoU is the average number of times the predicted value coincides with the actual value in each category. A greater value of MIoU indicates more accurate network prediction.

The FCN is commonly used for ship image semantic segmentation [37]. An FCN classifies images at a pixel level by selecting a sliding window for each pixel. Unet (unity networking), an improvement on the FCN [38], has been widely used in the field of transportation. Therefore, we compared these two models with DeepLabV3+. The IoU results for background and foreground after training for 50 epochs are shown in Table 4. It can be seen that DeepLabV3+ has clear advantages over FCN and Unet.

**Table 4.** Comparison test results of semantic segmentation.

|            | Background_IoU (%) | Boat_IoU (%) | Mean_IoU (%) |
| :--------: | :----------------: | :----------: | :----------: |
| FCN        | 99.42              | 79.80        | 89.61        |
| Unet       | 99.28              | 67.35        | 83.32        |
| DeeplabV3+ | **99.54**          | **82.61**    | **91.07**    |

The output images of DeepLabV3+ needed to be trimmed. Trimming was performed by calculating the number of pixels in the stern area along the transverse axis of the ship, and setting a threshold to eliminate scattered pixels to avoid individual misclassified pixels having undue impact on the corrected image. The segmentation results are shown in the first three columns of Figure 10. It can be seen from Figure 10 that the DeepLabV3+ model segmented the target area well without obstruction from the superstructure. The fourth column in Figure 10 shows the trimmed images that represent the actual ship contour region after coordinate mapping. The pixel size of the image corresponds to a real-world distance of 115 m. It is clear from these results that the size and heading of ships at different distances in the image were approximately estimated.

A comprehensive review of all the experimental results shows that the DeepLabV3+ semantic segmentation model can be successfully used on a high-performance server that receives automatic photos and uploading based on lightweight object detection and accomplishes contour extraction tasks on the cloud server, thus avoiding bandwidth usage for transmitting large amounts of remote video data. The object detection model will determine the rectangular region surrounding a vessel, increase the proportion of ship features in DeepLabV3+ input, and improve the pertinence of the semantic segmentation model.

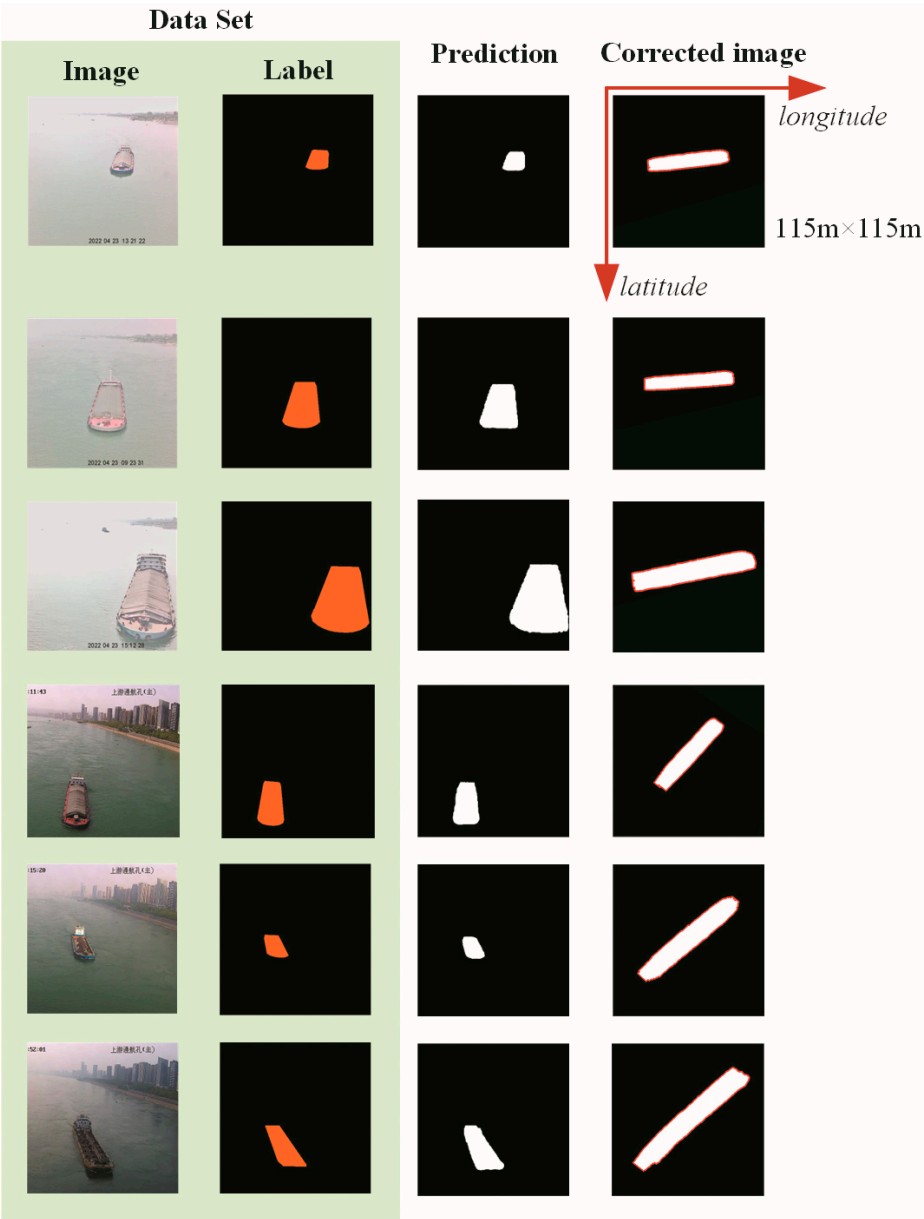

**Figure 10.** Ship contour extraction and size restoration results (The Non-English characters in the illustrations are meaningless).

We captured the video frame at the same time of receiving the AIS signal, and extracted the ship's aera with the semantic segmentation model. Based on the restored image, edge detection was combined with the Hough transform to further detect the minimum enclosing rectangle of the ship. We tested for several ships by estimating their dimensions, which we then compared with the actual dimensions; the results are shown in Table 5. We found that the average length relative error was 5.84%, and the average width relative error was 7.53%; these values are in a reasonable range. We also noted that the maximum length error and width error were 14.2% and 16.5%, respectively; this was because the color and texture features of the ship are not obvious enough to cause semantic segmentation errors, and were further amplified by image correction.

**Table 5.** Ship dimension data calculation results and errors.

| MMSI | Length (m) | | | Width (m) | | |
|---|---|---|---|---|---|---|
| | Actual | Calculated | Error (%) | Actual | Calculated | Error (%) |
| 413773165 | 87 | 90.3 | 3.8 | 14 | 14.6 | 4.3 |
| 413774959 | 87 | 84.2 | 3.2 | 15 | 16.3 | 8.7 |
| 413779378 | 90 | 96.7 | 7.4 | 15 | 17.0 | 13.3 |
| 413781326 | 106 | 110.6 | 4.3 | 17 | 18.2 | 7.1 |
| 413783151 | 107 | 115.2 | 7.7 | 16 | 17.1 | 6.9 |
| 413803847 | 90 | 96.8 | 7.6 | 15 | 15.4 | 2.7 |
| 413811188 | 100 | 97.5 | 2.5 | 16 | 16.9 | 5.6 |
| 413819165 | 100 | 114.2 | 14.2 | 17 | 19.8 | 16.5 |
| 413831856 | 110 | 113.6 | 3.3 | 19 | 20.0 | 5.3 |
| 413801536 | 107 | 103.1 | 3.6 | 16 | 16.6 | 3.8 |
| Mean | - | - | 5.8 | - | - | 7.4 |

According to the experimental results of Park et al. [4] using satellite-observed ships, the RMS errors for the length and width were 12.1 m and 6.8 m, respectively. We further calculated that the RMS error according to Table 5 and obtained the corresponding results as 6.6 m and 1.4 m, respectively, which are obviously better. This final result is valuable for creating and improving AIS data, and also provides a foundation for calculating ship collision risk.

## 5. Conclusions

Ship dimension information is important at the micro level in ship collision risk calculations. To avoid having to work with missing or incorrect ship dimension data, deep learning algorithms were used to extract ship contours from inland waterway surveillance video and real-time AIS information. According to the experimental results, the proposed object detection model and semantic segmentation model were very accurate in our experimental trials, and thus successfully resolved the missing AIS data issue.

The easily used lightweight ship object detection model that we developed for edge computing processed at a high frame rate without GPU acceleration, and facilitated the automatic acquisition and uploading of ship images. Therefore, it provides crucial support for shipping control on inland waterways.

In this study, AIS real-time position information was taken as a virtual control point to create a coordinate mapping model that rapidly and accurately mapped sensor image coordinates onto water surface coordinates without the need to rely on various projection parameters. Therefore, this method merits future development to further promote the combination of video, AIS, and radar information.

The combination of the object detection model and the coordinate mapping model, together with the use of the semantic segmentation algorithm, allowed us to extract the ship contour from the image and predict its actual size. The experimental trial results suggest the method is effective, and demonstrates the capability to extract ship dimension information automatically. This capability will benefit inland waterway regulatory authorities by enabling them to improve management of ship navigation.

The method still has some limitations. On the one hand, it has requirements regarding camera heights and shooting angles for coordinate mapping, as described in Section 3.2; therefore, a camera is best installed in the midspan of an inland river bridge with a bridge floor elevation of more than 50 m. On the other hand, the contour extraction method is not suitable for passenger ships and container ships, because their true contours are poorly reflected in the image.

Errors in ship contour extraction mainly come from the coordinate mapping model and semantic segmentation model. In our future research, we will add more ship images to the semantic segmentation dataset to improve the generalizability of the dataset so that the model can more exactly segment images of different types of ships. Vessel GPS

positioning generally exists within 1 m of error, which can affect the effectiveness of the coordinate mapping method. Thus, we will explore the error correction method based on constructing the trajectory equation to predict the GPS positioning coordinates according to the characteristics of GPS positioning coordinates obeying the Gaussian distribution at a certain moment. We will also endeavor to constantly use ships to simulate the control points during the experiment, in order to continuously update the optimal homography matrix.

**Author Contributions:** Methodology, validation, writing, original draft—Z.H. and Q.M.; data curation—L.L. and C.Y.; review and editing, funding acquisition—Q.H. All authors have read and agreed to the published version of the manuscript.

**Funding:** This research is supported by the National Natural Science Foundation of China (Grant No. 52372316).

**Institutional Review Board Statement:** Not applicable.

**Informed Consent Statement:** Informed consent was obtained from all subjects involved in the study.

**Data Availability Statement:** Not applicable.

**Conflicts of Interest:** The authors declare no conflict of interest.

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
