# Peer review of "Online Estimation of Ship Dimensions by Combining Images with AIS Reports"

_jmse, doi:10.3390/jmse11091700_

Round 1

Reviewer 1 Report

The paper presents a reconstruction of the ships' position and geometric features using a single camera and compares them with AIS data. Excellent, clear and well presented study area and result. Congratulations to the authors. What I may be missing is a discussion of camera position relative to ships in terms of height above water and angle to passing ships. What might be the critical camera positions that limit the method? This will open the discussion to the limitations of the method.

No comments here.

Reviewer 2 Report

Dear authors

Thank you for your research is very interesting. Because I have such a job, I had to find some mistakes hence please refer to the comments below in your work.

In line 188 on page 6 please explain word PAN, FCOS and maybe I suggested word nanodet maybe is better NaNodet because all word for AI algorithm has big letters.

In line 250 what means LC model?

Where is  Step2, and Eq. (3) is used to calculate the corresponding .. groups of water surface coordinates for p1 and p2?? Can you more explain?

Why you used initial learning rate 0.01?? This is this a heuristic or nominal value? Can you added one more sentence.

Please change name on figure 7 please write more because P-R curve is not enough.

Thank you for your research.

Reviewer 3 Report

The paper deals with an important and timely topic. The paper is nicely prepared, and I have only minor comments and suggestions given below.

1. The study's novelty should be clearly stated in the Introduction section.

2. Figure 3: The word "video" is miswritten as "vedio".

3. Figure 7. It is written "Presition"; shouldn't it be "precision"?

3. Lines 420-422 - It is stated: "We tested for several ships by estimating their dimensions which we then compared with the actual dimensions; the results are shown in Table 5". It is not stated how were the test ships chosen and which methodology was used. Please include.

4. In Table 5, there is one ship with a length error of 14.2% and a width error of 16.5%. Could you please elaborate more on this?

5. I suggest adding a discussion where the study's results could be compared with similar studies' results.

6. Are there any limitations of the study?

I hope that my comments and suggestions will be helpful.

In my opinion, the English Language is fine, only minor editing is required.

Reviewer 4 Report

see attachment
